# Benchmarking Multi-Agent Deep Reinforcement Learning Algorithms in Cooperative Tasks

**Georgios Papoudakis** *
School of Informatics
University of Edinburgh
g.papoudakis@ed.ac.uk

**Filippos Christianos** *
School of Informatics
University of Edinburgh
f.christianos@ed.ac.uk

**Lukas Schäfer**
School of Informatics
University of Edinburgh
l.schaefer@ed.ac.uk

**Stefano V. Albrecht**
School of Informatics
University of Edinburgh
s.albrecht@ed.ac.uk

## Abstract

Multi-agent deep reinforcement learning (MARL) suffers from a lack of commonly-used evaluation tasks and criteria, making comparisons between approaches difficult. In this work, we provide a systematic evaluation and comparison of three different classes of MARL algorithms (independent learning, centralised multi-agent policy gradient, value decomposition) in a diverse range of cooperative multi-agent learning tasks. Our experiments serve as a reference for the expected performance of algorithms across different learning tasks, and we provide insights regarding the effectiveness of different learning approaches. We open-source EPy-MARL, which extends the PyMARL codebase to include additional algorithms and allow for flexible configuration of algorithm implementation details such as parameter sharing. Finally, we open-source two environments for multi-agent research which focus on coordination under sparse rewards.

## 1 Introduction

Multi-agent reinforcement learning (MARL) algorithms use RL techniques to co-train a set of agents in a multi-agent system. Recent years have seen a plethora of new MARL algorithms which integrate deep learning techniques [Papoudakis et al., 2019, Hernandez-Leal et al., 2019]. However, comparison of MARL algorithms is difficult due to a lack of established benchmark tasks, evaluation protocols, and metrics. While several comparative studies exist for single-agent RL [Duan et al., 2016, Henderson et al., 2018, Wang et al., 2019], we are unaware of such comparative studies for recent MARL algorithms. Albrecht and Ramamoorthy [2012] compare several MARL algorithms but focus on the application of classic (non-deep) approaches in simple matrix games. Such comparisons are crucial in order to understand the relative strengths and limitations of algorithms, which may guide practical considerations and future research.

We contribute a **comprehensive empirical comparison of nine MARL algorithms in a diverse set of cooperative multi-agent tasks**. We compare three classes of MARL algorithms: independent learning, which applies single-agent RL algorithms for each agent without consideration of the multi-agent structure [Tan, 1993]; centralised multi-agent policy gradient [Lowe et al., 2017, Foerster et al., 2018, Yu et al., 2021]; and value decomposition [Sunehag et al., 2018, Rashid et al., 2018] algorithms. The two latter classes of algorithms follow the Centralised Training Decentralised

---

*Equal Contribution

Execution (CTDE) paradigm. These algorithm classes are frequently used in the literature either as baselines or building blocks for more complex algorithms [He et al., 2016, Sukhbaatar et al., 2016, Foerster et al., 2016, Raileanu et al., 2018, Jaques et al., 2019, Iqbal and Sha, 2019, Du et al., 2019, Ryu et al., 2020]. We evaluate algorithms in two matrix games and four multi-agent environments, in which we define a total of 25 different cooperative learning tasks. Hyperparameters of each algorithm are optimised separately in each environment using a grid-search, and we report the maximum and average evaluation returns during training. We run experiments with shared and non-shared parameters between agents, a common implementation detail in MARL that has been shown to affect converged returns [Christianos et al., 2021]. In addition to reporting detailed benchmark results, we analyse and discuss insights regarding the effectiveness of different learning approaches.

To facilitate our comparative evaluation, we created the open-source codebase **EPyMARL** (Extended PyMARL)[2], an extension of PyMARL [Samvelyan et al., 2019] which is commonly used in MARL research. EPyMARL implements additional algorithms and allows for flexible configuration of different implementation details, such as whether or not agents share network parameters. Moreover, we have implemented and open-sourced **two new multi-agent environments**: Level-Based Foraging (LBF) and Multi-Robot Warehouse (RWARE). With these environments we aim to test the algorithms' ability to learn coordination tasks under sparse rewards and partial observability.

## 2 Algorithms

### 2.1 Independent Learning (IL)

For IL, each agent is learning independently and perceives the other agents as part of the environment.

**IQL:** In Independent Q-Learning (IQL) [Tan, 1993], each agent has a decentralised state-action value function that is conditioned only on the local history of observations and actions of each agent. Each agent receives its local history of observations and updates the parameters of the Q-value network [Mnih et al., 2015] by minimising the standard Q-learning loss [Watkins and Dayan, 1992].

**IA2C:** Independent synchronous Advantage Actor-Critic (IA2C) is a variant of the commonly-used A2C algorithm [Mnih et al., 2016, Dhariwal et al., 2017] for decentralised training in multi-agent systems. Each agent has its own actor to approximate the policy and critic network to approximate the value-function. Both actor and critic are trained, conditioned on the history of local observations, actions and rewards the agent perceives, to minimise the A2C loss.

**IPPO:** Independent Proximal Policy Optimisation (IPPO) is a variant of the commonly-used PPO algorithm [Schulman et al., 2017] for decentralised training in multi-agent systems. The architecture of IPPO is identical to IA2C. The main difference between PPO and A2C is that PPO uses a surrogate objective which constrains the relative change of the policy at each update, allowing for more update epochs using the same batch of trajectories. In contrast to PPO, A2C can only perform one update epoch per batch of trajectories to ensure that the training batch remains on-policy.

### 2.2 Centralised Training Decentralised Execution (CTDE)

In contrast to IL, CTDE allows sharing of information during training, while policies are only conditioned on the agents' local observations enabling decentralised execution.

**Centralised policy gradient methods**   One category of CTDE algorithms are centralised policy gradient methods in which each agent consists of a decentralised actor and a centralised critic, which is optimised based on shared information between the agents.

**MADDPG:** Multi-Agent DDPG (MADDPG) [Lowe et al., 2017] is a variation of the DDPG algorithm [Lillicrap et al., 2015] for MARL. The actor is conditioned on the history of local observations, while critic is trained on the joint observation and action to approximate the joint state-action value function. Each agent individually minimises the deterministic policy gradient loss [Silver et al., 2014]. A common assumption of DDPG (and thus MADDPG) is differentiability of actions with respect to the parameters of the actor, so the action space must be continuous. Lowe et al. [2017] apply the Gumbel-Softmax trick [Jang et al., 2017, Maddison et al., 2017] to learn in discrete action spaces.

---

[2]`https://github.com/uoe-agents/epymarl`

Table 1: Overview of algorithms and their properties.

| | Centr. Training | Off-/On-policy | Value-based | Policy-based |
|---|---|---|---|---|
| IQL | ✗ | Off | ✓ | ✗ |
| IA2C | ✗ | On | ✓ | ✓ |
| IPPO | ✗ | On | ✓ | ✓ |
| MADDPG | ✓ | Off | ✓ | ✓ |
| COMA | ✓ | On | ✓ | ✓ |
| MAA2C | ✓ | On | ✓ | ✓ |
| MAPPO | ✓ | On | ✓ | ✓ |
| VDN | ✓ | Off | ✓ | ✗ |
| QMIX | ✓ | Off | ✓ | ✗ |

**COMA:** In Counterfactual Multi-Agent (COMA) Policy Gradient, Foerster et al. [2018] propose a modification of the advantage in the actor's loss computation to perform counterfactual reasoning for credit assignment in cooperative MARL. The advantage is defined as the discrepancy between the state-action value of the followed joint action and a counterfactual baseline. The latter is given by the expected value of each agent following its current policy while the actions of other agents are fixed. The standard policy loss with this modified advantage is used to train the actor and the critic is trained using the TD-lambda algorithm [Sutton, 1988].

**MAA2C:** Multi-Agent A2C (MAA2C) is an actor-critic algorithm in which the critic learns a joint state value function (in contrast, the critics in MADDPG and COMA are also conditioned on actions). It extends the existing on-policy actor-critic algorithm A2C by applying centralised critics conditioned on the state of the environment rather than the individual history of observations. It is often used as a baseline in MARL research and is sometimes referred to as Central-V, because it computes a centralised state value function. However, MAPPO also computes a centralised state value function, and in order to avoid confusion we refer to this algorithm as MAA2C.

**MAPPO:** Multi-Agent PPO (MAPPO) [Yu et al., 2021] is an actor-critic algorithm (extension of IPPO) in which the critic learns a joint state value function, similarly to MAA2C. In contrast to MAA2C, which can only perform one update epoch per training batch, MAPPO can utilise the same training batch of trajectories to perform several update epochs.

**Value Decomposition**   Another recent CTDE research direction is the decomposition of the joint state-action value function into individual state-action value functions.

**VDN:** Value Decomposition Networks (VDN) [Sunehag et al., 2018] aim to learn a linear decomposition of the joint Q-value. Each agent maintains a network to approximate its own state-action values. VDN decomposes the joint Q-value into the sum of individual Q-values. The joint state-action value function is trained using the standard DQN algorithm [Watkins and Dayan, 1992, Mnih et al., 2015]. During training, gradients of the joint TD loss flow backwards to the network of each agent.

**QMIX:** QMIX [Rashid et al., 2018] extends VDN to address a broader class of environments. To represent a more complex decomposition, a parameterised mixing network is introduced to compute the joint Q-value based on each agent's individual state-action value function. A requirement of the mixing function is that the optimal joint action, which maximises the joint Q-value, is the same as the combination of the individual actions maximising the Q-values of each agent. QMIX is trained to minimise the DQN loss and the gradient is backpropagated to the individual Q-values.

## 3   Multi-Agent Environments

We evaluate the algorithms in two finitely repeated matrix games and four multi-agent environments within which we define a total of 25 different learning tasks. All tasks are fully-cooperative, i.e. all agents receive identical reward signals. These tasks range over various properties including the degree of observability (whether agents can see the full environment state or only parts of it), reward density (receiving frequent/dense vs infrequent/sparse non-zero rewards), and the number of agents involved. Table 2 lists environments with properties, and we give brief descriptions below. We

Table 2: Overview of environments and properties.

|  | Observability | Rew. Sparsity | Agents | Main Difficulty |
|---|---|---|---|---|
| Matrix Games | Full | Dense | 2 | Sub-optimal equilibria |
| MPE | Partial / Full | Dense | 2-3 | Non-stationarity |
| SMAC | Partial | Dense | 2-10 | Large action space |
| LBF | Partial / Full | Sparse [3] | 2-4 | Coordination |
| RWARE | Partial | Sparse | 2-4 | Sparse reward |

believe each of the following environments addresses a specific challenge of MARL. Full details of environments and learning tasks are provided in Appendix C.

## 3.1 Repeated Matrix Games

We consider two cooperative matrix games proposed by Claus and Boutilier [1998]: the *climbing* and *penalty* game. The common-payoff matrices of the climbing and penalty game, respectively, are:

$$\begin{bmatrix} 0 & 6 & 5 \\ -30 & 7 & 0 \\ 11 & -30 & 0 \end{bmatrix} \qquad \begin{bmatrix} k & 0 & 10 \\ 0 & 2 & 0 \\ 10 & 0 & k \end{bmatrix}$$

where $k \leq 0$ is a penalty term. We evaluate in the penalty game for $k \in \{-100, -75, -50, -25, 0\}$. The difficulty of this game strongly correlates with $k$: the smaller $k$, the harder it becomes to identify the optimal policy due to the growing risk of penalty $k$. Both games are applied as repeated matrix games with an episode length of 25 and agents are given constant observations at each timestep. These matrix games are challenging due to the existence of local minima in the form of sub-optimal Nash equilibria [Nash, 1951]. Slight deviations from optimal policies by one of the agents can result in significant penalties, so agents might get stuck in risk-free (deviations from any agent does not significantly impede payoff) local optima.

## 3.2 Multi-Agent Particle Environment

The Multi-Agent Particle Environments (MPE) [Mordatch and Abbeel, 2017] consists of several two-dimensional navigation tasks. We investigate four tasks that emphasise coordination: Speaker-Listener, Spread, Adversary[4], and Predator-Prey[4]. Agent observations consist of high-level feature vectors including relative agent and landmark locations. The actions allow for two-dimensional navigation. All tasks but Speaker-Listener, which also requires binary communication, are fully observable. MPE tasks serve as a benchmark for agent coordination and their ability to deal with non-stationarity [Papoudakis et al., 2019] due to significant dependency of the reward with respect to joint actions. Individual agents not coordinating effectively can severely reduce received rewards.

## 3.3 StarCraft Multi-Agent Challenge

The StarCraft Multi-Agent Challenge (SMAC) [Samvelyan et al., 2019] simulates battle scenarios in which a team of controlled agents must destroy an enemy team using fixed policies. Agents observe other units within a fixed radius, and can move around and select enemies to attack. We consider five tasks in this environment which vary in the number and types of units controlled by agents. The primary challenge within these tasks is the agents' ability to accurately estimate the value of the current state under partial observability and a growing number of agents of diverse types across tasks. Latter leads to large action spaces for agents which are able to select other agents or enemy units as targets for healing or attack actions, respectively, depending on the controlled unit.

---

[3]Rewards in LBF are sparser compared to MPE and SMAC, but not as sparse as in RWARE.

[4]Adversary and Predator-Prey are originally competitive tasks. The agents controlling the adversary and prey, respectively, are controlled by a pretrained policy obtained by training all agents with the MADDPG algorithm for 25000 episodes (see Appendix C for details).

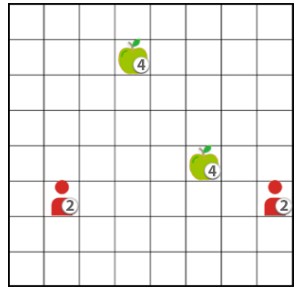
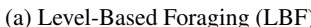

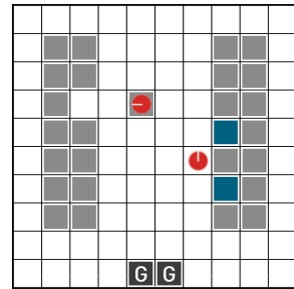

(a) Level-Based Foraging (LBF)     (b) Multi-Robot Warehouse (RWARE)

Figure 1: Illustrations of the open-sourced multi-agent environments [Christianos et al., 2020].

### 3.4 Level-Based Foraging

In Level-Based Foraging (LBF) [Albrecht and Ramamoorthy, 2013, Albrecht and Stone, 2017] agents must collect food items which are scattered randomly in a grid-world. Agents and items are assigned levels, such that a group of one or more agents can collect an item if the sum of their levels is greater or equal to the item's level. Agents can move in four directions, and have an action that attempts to load an adjacent item (the action will succeed depending on the levels of agents attempting to load the particular item). LBF allows for many different tasks to be configured, including partial observability or a highly cooperative task where all agents must simultaneously participate to collect the items. We define seven distinct tasks with a variable world size, number of agents, observability, and cooperation settings indicating whether all agents are required to load a food item or not. We implemented the LBF environment which is publicly available on GitHub, under the MIT licence: `https://github.com/uoe-agents/lb-foraging`.

### 3.5 Multi-Robot Warehouse

The Multi-Robot Warehouse environment (RWARE) represents a cooperative, partially-observable environment with sparse rewards. RWARE simulates a grid-world warehouse in which agents (robots) must locate and deliver requested shelves to workstations and return them after delivery. Agents are only rewarded for completely delivering requested shelves and observe a $3 \times 3$ grid containing information about the surrounding agents and shelves. The agents can move forward, rotate in either direction, and load/unload a shelf. We define three tasks which vary in world size, number of agents and shelf requests. The sparsity of the rewards makes this a hard environment, since agents must correctly complete a series of actions before receiving any reward. Additionally, observations are sparse and high-dimensional compared to the other environments. RWARE is the second environment we designed and open-source under the MIT licence: `https://github.com/uoe-agents/robotic-warehouse`.

We have developed and plan to maintain the LBF and RWARE environments as part of this work. They have already been used in other multi-agent research [Christianos et al., 2020, 2021, Rahman et al., 2021, Papoudakis et al., 2021]. For more information including installation instructions, interface guides with code snippets and detailed descriptions, see Appendix A.

## 4 Evaluation

### 4.1 Evaluation Protocol

To account for the improved sample efficiency of off-policy over on-policy algorithms and to allow for fair comparisons, we train on-policy algorithms for a factor of ten more samples than off-policy algorithms. In MPE and LBF we train on-policy algorithms for 20 million timesteps and off-policy algorithms for two million timesteps, while in SMAC and RWARE, we train on-policy and off-policy algorithms for 40 and four million timesteps, respectively. By not reusing samples through an experience replay buffer, on-policy algorithms are less sample efficient, but not generally slower (if the simulator is reasonably fast) and thus this empirical adjustment is fair. We perform in total 41 evaluations of each algorithm at constant timestep intervals during training, and at each evaluation

point we evaluate for 100 episodes. In matrix games, we train off-policy algorithms for 250 thousand timesteps and on-policy algorithms for 2.5 million timesteps and evaluate every 2.5 thousand and 25 thousand timesteps, respectively, for a total of 100 evaluations.

## 4.2 Parameter Sharing

Two common configurations for training deep MARL algorithms are: without and with parameter sharing. Without parameter sharing, each agent uses its own set of parameters for its networks. Under parameter sharing, all agents share the same set of parameters for their networks. In the case of parameter sharing, the policy and critic (if there is one) additionally receive the identity of each agent as a one-hot vector input. This allows for each agent to develop a different behaviour. The loss is computed over all agents and used to optimise the shared parameters. In the case of varying input sizes across agents, inputs are zero-padded to ensure identical input dimensionality. Similarly if agents have varying numbers of actions, action selection probabilities for invalid actions are set to 0.

## 4.3 Hyperparameter Optimisation

Hyperparameter optimisation was performed for each algorithm separately in each environment. From each environment, we selected one task and optimised the hyperparameters of all algorithms in this task. In the MPE environment, we perform the hyperparameter optimisation in the Speaker-Listener task, in the SMAC environment in the "3s5z" task, in the LBF environment in the "15x15-3p-5f" task, and in the RWARE environment in the "Tiny 4p" task. We train each combination of hyperparameters using three different seeds and compare the maximum evaluation returns. The best performing combination on each task is used for all tasks in the respective environment for the final experiments. In Appendix I, we present the hyperparameters that were used in each environment and algorithm.

## 4.4 Performance Metrics

**Maximum returns:** For each algorithm, we identify the evaluation timestep during training in which the algorithm achieves the highest average evaluation returns across five random seeds. We report the average returns and the 95% confidence interval across five seeds from this evaluation timestep.

**Average returns:** We also report the average returns achieved throughout all evaluations during training. Due to this metric being computed over all evaluations executed during training, it considers learning speed besides final achieved returns.

## 4.5 Computational Requirements

All experiments presented in this work were executed purely on CPUs. The experiments were executed in compute clusters that consist of several nodes. The main types of CPU models that were used for this work are Intel(R) Xeon(R) CPU E5-2630 v3 @ 2.40GHz and AMD EPYC 7502 32-Core processors. All but the SMAC experiments were executed using a single CPU core. All SMAC experiments were executed using 5 CPU cores. The total number of CPU hours that were spent for executing the experiments in this work (excluding the hyperparameter search) are 138,916.

## 4.6 Extended PyMARL

Implementation details in reinforcement learning significantly affect the returns that each algorithm achieves [Andrychowicz et al., 2021]. To enable consistent evaluation of MARL algorithms, we open-source the Extended PyMARL (EPyMARL) codebase. EPyMARL is an extension of the PyMARL codebase [Samvelyan et al., 2019]. PyMARL provides implementations for IQL, COMA, VDN and QMIX. We increase the scope of the codebase to include five additional policy gradients algorithms: IA2C, IPPO, MADDPG, MAA2C and MAPPO. The original PyMARL codebase implementation assumes that agents share parameters and that all the agents' observation have the same shape. In general, parameter sharing is a commonly applied technique in MARL. However, it was shown that parameter sharing can act as an information bottleneck, especially in environments with heterogeneous agents [Christianos et al., 2021]. EPyMARL allows training MARL algorithms without parameter sharing, training agents with observations of varying dimensionality, and tuning several implementation details such as reward standardisation, entropy regularisation, and the use of

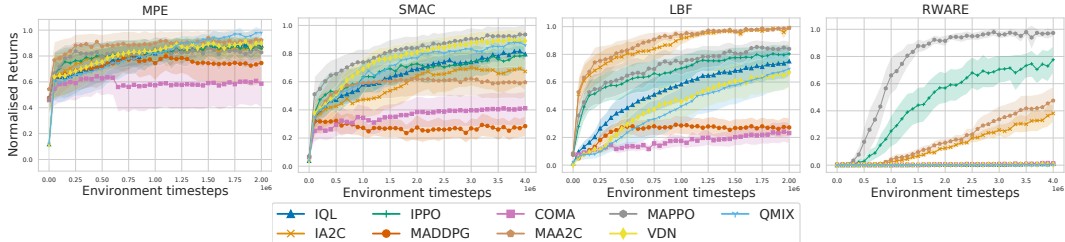

Figure 2: Normalised evaluation returns averaged over the tasks in the all environments except matrix games. Shadowed part represents the 95% confidence interval.

Table 3: Maximum returns and 95% confidence interval over five seeds for all nine algorithms with parameter sharing in all 25 tasks. The highest value in each task is presented in bold. Asterisks denote the algorithms that are not significantly different from the best performing algorithm in each task.

| Tasks \Algs. | IQL | IA2C | IPPO | MADDPG | COMA | MAA2C | MAPPO | VDN | QMIX |
|---|---|---|---|---|---|---|---|---|---|
| **Matrix Games** | | | | | | | | | |
| Climbing | **195.00 ± 67.82** | 175.00 ± 0.00 | 175.00 ± 0.00 | 170.00 ± 10.00 | 185.00 ± 48.99 | 175.00 ± 0.00 | 175.00 ± 0.00 | 175.00 ± 54.77 | 175.00 ± 54.77 |
| Penalty k=0 | **250.00 ± 0.00** | **250.00 ± 0.00** | **250.00 ± 0.00** | 249.98 ± 0.04 | **250.00 ± 0.00** | **250.00 ± 0.00** | **250.00 ± 0.00** | **250.00 ± 0.00** | **250.00 ± 0.00** |
| Penalty k=-25 | 50.00 ± 0.00 | 50.00 ± 0.00 | 50.00 ± 0.00 | 49.97 ± 0.02 | 50.00 ± 0.00 | 50.00 ± 0.00 | 50.00 ± 0.00 | 50.00 ± 0.00 | 50.00 ± 0.00 |
| Penalty k=-50 | 50.00 ± 0.00 | 50.00 ± 0.00 | 50.00 ± 0.00 | 49.98 ± 0.02 | 50.00 ± 0.00 | 50.00 ± 0.00 | 50.00 ± 0.00 | 50.00 ± 0.00 | 50.00 ± 0.00 |
| Penalty k=-75 | 50.00 ± 0.00 | 50.00 ± 0.00 | 50.00 ± 0.00 | 49.97 ± 0.02 | 50.00 ± 0.00 | 50.00 ± 0.00 | 50.00 ± 0.00 | 50.00 ± 0.00 | 50.00 ± 0.00 |
| Penalty k=-100 | 50.00 ± 0.00 | 50.00 ± 0.00 | 50.00 ± 0.00 | 49.97 ± 0.03 | 50.00 ± 0.00 | 50.00 ± 0.00 | 50.00 ± 0.00 | 50.00 ± 0.00 | 50.00 ± 0.00 |
| **MPE** | | | | | | | | | |
| Speaker-Listener | −18.36 ± 4.67 | −12.60 ± 3.62 * | −13.10 ± 3.50 | −13.56 ± 1.73 | −30.40 ± 5.18 | −10.71 ± 0.38 * | **−10.68 ± 0.30** | −15.95 ± 2.48 | −11.56 ± 0.53 |
| Spread | −132.63 ± 2.22 | −134.43 ± 1.15 | −133.86 ± 3.67 | −141.70 ± 1.74 | −204.31 ± 6.30 | −129.90 ± 1.63 * | −133.54 ± 3.08 | −131.03 ± 1.85 | **−126.62 ± 2.96** |
| Adversary | 9.38 ± 0.91 | 12.12 ± 0.44 * | **12.17 ± 0.32** | 8.97 ± 0.89 | 8.05 ± 0.89 | 12.06 ± 0.45 * | 11.30 ± 0.38 | 9.28 ± 0.90 | 9.67 ± 0.66 |
| Tag | 22.18 ± 2.83 | 17.44 ± 1.31 | 19.44 ± 2.94 | 12.50 ± 6.30 | 8.72 ± 4.42 | 19.95 ± 7.15 * | 18.52 ± 5.64 | 24.50 ± 2.19 | **31.18 ± 3.81** |
| **SMAC** | | | | | | | | | |
| 2s_vs_1sc | 16.72 ± 0.38 | 20.24 ± 0.00 | 20.24 ± 0.01 | 13.14 ± 2.01 | 11.04 ± 7.21 | 20.20 ± 0.05 * | **20.25 ± 0.00** | 18.04 ± 0.33 | 19.01 ± 0.40 |
| 3s5z | 16.44 ± 0.15 | 18.56 ± 1.31 * | 13.36 ± 2.08 | 12.04 ± 0.82 | 18.90 ± 1.01 * | 19.95 ± 0.05 * | **20.39 ± 1.14** | 19.57 ± 0.20 * | 19.66 ± 0.14 * |
| corridor | 15.72 ± 1.77 | **18.59 ± 0.62** | 17.97 ± 3.44 * | 5.85 ± 0.58 | 7.75 ± 0.19 | 8.97 ± 0.29 | 17.14 ± 4.39 * | 15.25 ± 4.18 * | 16.45 ± 3.54 * |
| MMM2 | 13.69 ± 1.02 | 10.70 ± 2.77 | 11.37 ± 1.15 | 3.96 ± 0.32 | 6.95 ± 0.27 | 10.37 ± 1.95 | 17.78 ± 0.44 | **18.49 ± 0.31** | 18.40 ± 0.24 * |
| 3s_vs_5z | **21.15 ± 0.41** | 4.42 ± 0.02 | 19.36 ± 6.15 * | 5.99 ± 0.58 | 3.23 ± 0.05 | 6.68 ± 0.55 | 18.17 ± 4.17 * | 19.03 ± 5.77 * | 16.04 ± 2.87 |
| **LBF** | | | | | | | | | |
| 8x8-2p-2f-c | **1.00 ± 0.00** | **1.00 ± 0.00** | **1.00 ± 0.00** | 0.46 ± 0.02 | 0.61 ± 0.30 | **1.00 ± 0.00** | **1.00 ± 0.00** | **1.00 ± 0.00** | 0.96 ± 0.07 * |
| 8x8-2p-2f-2s-c | **1.00 ± 0.00** | **1.00 ± 0.00** | 0.78 ± 0.05 | 0.70 ± 0.04 | 0.45 ± 0.15 | **1.00 ± 0.00** | 0.85 ± 0.06 | **1.00 ± 0.00** | **1.00 ± 0.00** |
| 10x10-3p-3f | 0.93 ± 0.02 | **1.00 ± 0.00** | 0.98 ± 0.01 | 0.24 ± 0.04 | 0.19 ± 0.06 | **1.00 ± 0.00** | 0.99 ± 0.01 | 0.84 ± 0.08 | 0.84 ± 0.08 |
| 10x10-3p-3f-2s | 0.86 ± 0.01 | 0.94 ± 0.03 * | 0.70 ± 0.03 | 0.41 ± 0.03 | 0.29 ± 0.12 | **0.96 ± 0.02** | 0.72 ± 0.03 | 0.90 ± 0.03 | 0.90 ± 0.01 |
| 15x15-3p-5f | 0.17 ± 0.08 | **0.89 ± 0.04** | 0.77 ± 0.08 | 0.10 ± 0.02 | 0.08 ± 0.04 | 0.87 ± 0.06 * | 0.77 ± 0.02 | 0.15 ± 0.02 | 0.09 ± 0.04 |
| 15x15-4p-3f | 0.54 ± 0.18 | 0.99 ± 0.01 * | 0.98 ± 0.01 | 0.17 ± 0.03 | 0.17 ± 0.04 | **1.00 ± 0.00** | 0.96 ± 0.02 | 0.38 ± 0.13 | 0.15 ± 0.06 |
| 15x15-4p-5f | 0.22 ± 0.04 | 0.93 ± 0.03 * | 0.67 ± 0.22 | 0.12 ± 0.06 | 0.12 ± 0.06 | **0.95 ± 0.01** | 0.70 ± 0.25 * | 0.30 ± 0.04 | 0.25 ± 0.09 |
| **RWARE** | | | | | | | | | |
| Tiny 4p | 0.72 ± 0.37 | 26.34 ± 4.60 | 31.82 ± 10.71 | 0.54 ± 0.10 | 1.16 ± 0.15 | 32.50 ± 9.79 | **49.42 ± 1.22** | 0.80 ± 0.28 | 0.30 ± 0.19 |
| Small 4p | 0.14 ± 0.28 | 6.54 ± 1.15 | 19.78 ± 3.12 | 0.18 ± 0.12 | 0.16 ± 0.16 | 10.30 ± 1.48 | **27.00 ± 1.80** | 0.18 ± 0.27 | 0.06 ± 0.08 |
| Tiny 2p | 0.28 ± 0.38 | 8.18 ± 1.25 | 20.22 ± 1.76 * | 0.44 ± 0.34 | 0.48 ± 0.34 | 8.38 ± 2.59 | **21.16 ± 1.50** | 0.12 ± 0.07 | 0.14 ± 0.19 |

recurrent or fully-connected networks. EPyMARL is publicly available on GitHub and distributed under the Apache License: `https://github.com/uoe-agents/epymarl`.

## 5   Results

In this section we compile the results across all environments and algorithms. Figure 2 presents the normalised evaluation returns in all environments, except matrix games. We normalise the returns of all algorithms in each task in the $[0, 1]$ range using the following formula: $\text{norm\_}G_t^a = (G_t^a − \min(G_t))/(\max(G_t) − \min(G_t))$, where $G_t^a$ is the return of algorithm $a$ in task $t$, and $G_t$ is the returns of all algorithms in task $t$. Table 3 presents the maximum returns for the nine algorithms in all 25 tasks with parameter sharing. The maximum returns without parameter sharing, as well as the average returns both with and without parameter sharing are presented in Appendix E. In Table 3, we highlight the highest mean in bold. We performed two-sided t-tests with a significance threshold of $0.05$ between the highest performing algorithm and each other algorithm in each task. If an algorithm's performance was *not* statistically significantly different from the best algorithm, the respective value is annotated with an asterisk (i.e. bold or asterisks in the table show the best performing algorithms per task). In the SMAC tasks, it is a common practice in the literature to report the win-rate as a percentage and not the achieved returns. However, we found it more informative to report the achieved returns since it is the metric that the algorithms aim to optimise. Moreover, higher returns do not always correspond to higher win-rates which can make the interpretation of the performance metrics more difficult. For completeness, we report the win-rates achieved by all algorithms in Appendix G.

## 5.1 Independent Learning

We find that IL algorithms perform adequately in all tasks despite their simplicity. However, performance of IL is limited in partially observable SMAC and RWARE tasks, compared to their CTDE counterparts, due to IL algorithms' inability to reason over joint information of agents.

**IQL:** IQL performs significantly worse than the other IL algorithms in the partially-observable Speaker-Listener task and in all RWARE tasks. IQL is particularly effective in all but three LBF tasks, where relatively larger grid-worlds are used. IQL achieves the best performance among all algorithms in the "3s_vs_5z" task, while it performs competitively in the rest of the SMAC tasks.

**IA2C:** The stochastic policy of IA2C appears to be particularly effective on all environments except in a few SMAC tasks. In the majority of tasks, it performs similarly to IPPO with the exception of RWARE and some SMAC tasks. However, it achieves higher returns than IQL in all but two SMAC tasks. Despite its simplicity, IA2C performs competitively compared to all CTDE algorithms, and significantly outperforms COMA and MADDPG in the majority of the tasks.

**IPPO:** IPPO in general performs competitively in all tasks across the different environments. On average (Figure 2) it achieves higher returns than IA2C in MPE, SMAC and RWARE tasks, but lower returns in the LBF tasks. IPPO also outperforms MAA2C in the partially-observable RWARE tasks, but in general it performs worse compared to its centralised MAPPO version.

## 5.2 Centralised Training Decentralised Execution

Centralised training aims to learn powerful critics over joint observations and actions to enable reasoning over a larger information space. We find that learning such critics is valuable in tasks which require significant coordination under partial observability, such as the MPE Speaker-Listener and harder SMAC tasks. In contrast, IL is competitive compared to CTDE algorithms in fully-observable tasks of MPE and LBF. Our results also indicate that in most RWARE tasks, MAA2C and MAPPO significantly improve the achieved returns compared to their IL (IA2C and IPPO) versions. However, training state-action value functions appears challenging in RWARE tasks with sparse rewards, leading to very low performance of the remaining CTDE algorithms (COMA, VDN and QMIX).

**Centralised Multi-Agent Policy Gradient**  Centralised policy gradient methods vary significantly in performance.

**MADDPG:** MADDPG performs worse than all the other algorithms except COMA, in the majority of the tasks. It only performs competitively in some MPE tasks. It also exhibits very low returns in discrete grid-world environments LBF and RWARE. We believe that these results are a direct consequence of the biased categorical reparametarisation using Gumbel-Softmax.

**COMA:** In general, COMA exhibits one of the lowest performances in most tasks and only performs competitively in one SMAC task. We found that COMA suffers very high variance in the computation of the counterfactual advantage. In the Speaker-Listener task, it fails to find the sub-optimal local minima solution that correspond to returns around to -17. Additionally, it does not exhibit any learning in the RWARE tasks in contrast to other on-policy algorithms.

**MAA2C:** MAA2C in general performs competitively in the majority of the tasks, except a couple of SMAC tasks. Compared to MAPPO, MAA2C achieves slightly higher returns in the MPE and the LBF tasks, but in most cases significantly lower returns in the SMAC and RWARE tasks.

**MAPPO:** MAPPO achieves high returns in the vast majority of tasks and only performs slightly worse than other algorithms in some MPE and LBF tasks. Its main advantage is the combination of on-policy optimisation with its surrogate objective which significantly improves the sample efficiency compared to MAA2C. Its benefits can be observed in RWARE tasks where its achieved returns exceed the returns of all other algorithms (but not always significantly).

**Value Decomposition**  Value decomposition is an effective approach in most environments. In the majority of tasks across all environments except RWARE, VDN and QMIX outperform or at least match the highest returns of any other algorithm. This suggests that VDN and QMIX share the major advantages of centralised training. In RWARE, VDN and QMIX do not exhibit any learning, similar to IQL, COMA and MADDPG, indicating that value decomposition methods require sufficiently dense rewards to successfully learn to decompose the value function into the individual agents.

**VDN:** While VDN and QMIX perform similarly in most environments, the difference in performance is most noticeable in some MPE tasks. It appears VDN's assumption of linear value function decomposition is mostly violated in this environment. In contrast, VDN and QMIX perform similarly in most SMAC tasks and across all LBF tasks, where the global utility can apparently be represented by a linear function of individual agents' utilities.

**QMIX:** Across almost all tasks, QMIX achieves consistently high returns, but does not necessarily achieve the highest returns among all algorithms. Its value function decomposition allows QMIX to achieve slightly higher returns in some of the more complicated tasks where the linear value decomposition of VDN in is not sufficient.

### 5.3 Parameter Sharing

Figure 3 presents the normalised maximum returns averaged over the nine algorithms and tasks with and without parameter sharing in all environments. We observe that in all environments except the matrix games, parameter sharing improves the returns over no parameter sharing. While the average values presented in Figure 3 do not seem statistically significant, by looking closer in Tables 3 and 7 we observe that in several cases of algorithm-task pairs the improvement due to parameter sharing seems significant. Such improvements can be observed for most algorithms in MPE tasks, especially in Speaker-Listener and Tag. For SMAC, we observe that parameter sharing improves the returns in harder tasks. Similar observations can be made for LBF and RWARE. In these environments, the return improvement of parameter sharing appears to correlate with the sparsity of rewards. For tasks with larger grid-worlds or fewer agents, where the reward is more sparse, parameter sharing

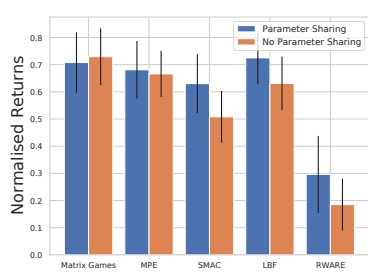

Figure 3: Normalised maximum returns averaged over all algorithms with/without parameter sharing (with standard error).

leads to large increases in returns compared to simpler tasks. This does not come as a surprise since parameter sharing uses a larger number of trajectories to train the same shared architecture to improve sample efficiency compared to no parameter sharing.

## 6 Analysis

**Independent learning can be effective in multi-agent systems. Why and when?** It is often stated that IL is inferior to centralised training methods due to the environment becoming non-stationary from the perspective of any individual agent. This is true in many cases and particularly crucial when IL is paired with off-policy training from an experience replay buffer, as pointed out by Lowe et al. [2017]. In our experiments, IQL trains agents independently using such a replay buffer and is thereby limited in its performance in tasks that require extensive coordination among the agents. There, agents depend on the information about other agents and their current behaviour to choose well-coordinated actions and hence learning such policies from a replay buffer (where other agents differ in their behaviour) appears infeasible. However, this is not a concern to multi-agent environments in general. In smaller SMAC tasks and most LBF tasks, each agent can learn a policy that achieves relatively high returns by utilising only its local observation history, and without requiring extensive coordination with other agents. E.g. in LBF, agents "only" have to learn to pick up the food until it is collected. Of course, they will have to coordinate such behaviour with other agents, but naively going to food (especially when others are also close) and attempting to pick it up can be a viable local minima policy, and hard to improve upon. Whenever more complicated coordination is required, such as simultaneously picking up an item with higher level, exploring and learning those joint actions becomes difficult. IA2C and IPPO on the other hand learn on-policy, so there is no such training from a replay buffer. These algorithms should be expected to achieve higher returns in the majority of the tasks as they learn given the currently demonstrated behaviour of other agents. As long as effective behaviour is eventually identified through time and exploration, IA2C and IPPO can learn more effective policies than IQL despite also learning independently.

**Centralised information is required for extensive coordination under partial-observability.** We note that the availability of joint information (observations and actions) over all agents serves as a

powerful training signal to optimise individual policies whenever the full state is not available to individual agents. Comparing the performance in Table 3 of IA2C and MAA2C, two almost identical algorithms aside from their critics, we notice that MAA2C achieves equal or higher returns in the majority of the tasks. This difference is particularly significant in tasks where agent observations lack important information about other agents or parts of the environment outside of their receptive field due to partial observability. This can be observed in RWARE tasks with 4 agents which require extensive coordination so that agents are not stuck in the narrow passages. However, in RWARE Tiny 2p task, the performance of IA2C and MAA2C is similar as only two agents rarely get stuck in the narrow passages. Finally, IA2C and MAA2C have access to the same information in fully-observable tasks, such as most LBF and MPE tasks, leading to similar returns. A similar pattern can be observed for IPPO and MAPPO. However, we also observe that centralised training algorithms such as COMA, MADDPG, VDN and QMIX are unable to learn effective behaviour in the partially-observable RWARE. We hypothesise that training larger networks over the joint observation- and action-space, as required for these algorithms, demands sufficient training signals. However, rewards are sparse in RWARE and observations are comparably large.

**Value decomposition – VDN vs QMIX.** Lastly, we address the differences observed in value decomposition applied by VDN and QMIX. Such decomposition offers an improvement in comparison to the otherwise similar IQL algorithm across most tasks. Both VDN and QMIX are different in their decomposition. QMIX uses a trainable mixing network to compute the joint Q-value compared to VDN which assumes linear decomposition. With the additional flexibility, QMIX aims to improve learnability, i.e. it simplifies the learning objective for each agent to maximise, while ensuring the global objective is maximised by all agents [Agogino and Tumer, 2008]. Such flexibility appears to mostly benefit convergence in harder MPE tasks, such as Speaker-Listener and Tag, but comes at additional expense seen in environments like LBF, where the decomposition did not have to be complex. It appears that the dependency of rewards with respect to complicated interactions between agents in MPE tasks and the resulting non-stationarity benefits more complex decomposition. Finally, VDN and QMIX perform significantly worse than the policy gradient methods (except COMA) in the sparse-reward RWARE tasks. This does not come as a surprise, since the utility of the agents is rarely greater than 0, which makes it hard to successfully learn the individual utilities.

## 7  Conclusion

We evaluated nine MARL algorithms in a total of 25 cooperative learning tasks, including combinations of partial/full observability, sparse/dense rewards, and a number of agents ranging from two to ten. We compared algorithm performance in terms of maximum and average returns. Additionally, we further analysed the effectiveness of independent learning, centralised training, and value decomposition and identify types of environments in which each strategy is expected to perform well. We created EPyMARL, an open-source codebase for consistent evaluation of MARL algorithms in cooperative tasks. Finally, we implement and open-source LBF and RWARE, two new multi-agent environments which focus on sparse-reward exploration which previous environments do not cover. Our work is limited to cooperative environments and commonly-used MARL algorithms. Competitive environments as well as solutions to a variety of MARL challenges such as exploration, communication, and opponent modelling require additional studies in the future. We hope that our work sheds some light on the relative strengths and limitations of existing MARL algorithms, to provide guidance in terms of practical considerations and future research.

## Funding Disclosure

This research was in part financially supported by the UK EPSRC Centre for Doctoral Training in Robotics and Autonomous Systems (G.P., F.C.), and the Edinburgh University Principal's Career Development Scholarship (L.S.).

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
