# OpenReview forum: "Benchmarking Multi-Agent Deep Reinforcement Learning Algorithms in Cooperative Tasks"
_NeurIPS.cc/2021/Track/Datasets_and_Benchmarks/Round1 — NeurIPS 2021 Datasets and Benchmarks Track (Round 1)_

### Official Review · Reviewer_BhPG · 2021-07-05
**A review paper or a benchmarking paper?**

**Rating:** 5
**Confidence:** 4
**Correctness:** Authors present data and analysis to …
**Clarity:** Paper is easy to follow.

**Strengths:**

- Comprehensive overview of the problem space and existing algorithms
- Open source codebase
- Well timed paper consolidating progress in a rapidly evolving field to facilitate fair comparisons
- A good collection of task suits from different environments
- Authors enlist their computational foot print


**Weaknesses:**

- My primary concern with the paper its positioning as a benchmarks paper. The paper to a large extent is a literature review summarizing ideas from the field into a single space. Both task (except two) and algorithms are rehash of previously published works. I'm fairly convinced about the contributions of the paper but I'm conflicted if this should instead be restructured as a review paper. Reason being -- review paper is underlined as a summary of other's contributions, instead of authors' contributions.
- Authors do propose the RWARE environments but Im not convinced if that is contribution enough to warrant a publication. Additionally  there is less evidence / experiments to support if these new environments are well structured, calibrated, and indeed useful to the community at large.
- Section 4.1: Under the premise of a fast simulator, authors propose adjustment for on-policy and off-policy methods. The goal of algorithmic developments are to make progress in real world where cost of sample acquisition is significant. (even for sim2real paradigm, compute cost is significant for non-grid world problems). I'm not convinced that this adjustment is adequate and justified.
- Authors largely rehash existing ideas of the field but miss on the opportunity to cater to the future needs (see Additional Feedback section for details)

**Additional Feedback:**

While the broader field of Robotics and Deep RL have seen significant success in the complexity of tasks that can be solved, MARL seems to be suck in grid world and simple settings. Real world is neither of those. Real world problems are complex with various nuances that are being ignored. Authors are very positioned to design a benchmark for the future needs of the MARL field. A few suggestions below

- Move beyond grid world and matrix games settings
- Device problems using real world data -- e.g. financial-markets
- Introduce heterogenous players -- players in a team rarely have same responsibilities and rewards
- Introduce co-operative and competitive settings
- Introduce continuous control problem -- Mujoco problems in openAIGym, dm_control [Emergent Complexity via Multi-Agent Competition. Trapit et al.]
- Team sports are great settings - e.g. football [From Motor Control to Team Play in Simulated Humanoid Football. siqi et al.]



**Documentation:**

Open source code provided

**Ethics:**

Minimal ethical concerns -- simulated setting with no real world data.

**Relation To Prior Work:**

Authors are candid about relation to prior works. They appropriately point relevance and contrast their contributions to existing literature.

**Summary And Contributions:**

- Efforts towards benchmarking Multi Agent Deep Reinforcement Learning (MARL)
- Compare and contrasts 3 different classes of MARL on proposed benchmarks
- Open source codebase with new environments

---

> ### Author Response · Authors · 2021-07-14
> **Response to reviewer BhPG**
>
> We would like to thank reviewer BhPG for their review.
>
> [W1] Indeed, we summarise recent state-of-the-art MARL algorithms and use some of the most commonly used environments. Our goal is to provide a comprehensive comparison that the community can benefit from. To that end, not only we propose/maintain two more environments, but have implemented the algorithms in a consistent manner to provide for a useful comparison. Unfortunately, algorithms such as IPPO or MAA2C are known to the community but their implementations are rarely consistent, and we tried to also address this.
>
> [W2] We point the reviewer to the last paragraph of Section 3: these environments have already been proven to be interesting for MARL problems by our tangential work [1,2,3].
>
> [W3] Two of the four environments are not grid-worlds: SMAC and MPE. SMAC can even be considered a real-world problem on its own since applications on video games can lead to advances in both the entertainment and educational fields.
> Also, we opted to adjust for off-policy and on-policy settings because there is use for on-policy algorithms and they should not be discarded only for their sample efficiency. In several of our experiments, on-policy algorithms could run faster (wall-clock time) than off-policy. There are several real-world use cases for multi-agent learning for which sample acquisition is insignificant compared to applications in the physical world:  multi-agent based authentication systems [4], video games, recommendation systems, and more [also please see the last paragraph].
>
> [AF1] Please see [W3] above. We already include 2 environments that are neither matrix nor grid-world games.
>
> [AF3] In cooperative tasks, all agents receive the same reward signals. However, SMAC does have heterogeneous agents controlling units of different abilities and action spaces.
>
> [AF4] This paper focuses on MARL algorithms evaluated in cooperative environments. Competitive environments are of course very interesting but they are not within the scope of this work. Research on cooperative settings often focuses on aspects that can not be applied in competitive environments. Broadening the scope of our research would lead to a convoluted paper. We believe a narrower scope allows for a clearer and comprehensible analysis.
>
> [AF5/6]  We believe that the selection of environments offers a lot of value, without being overly computationally expensive. They directly test agent interaction, credit assignment, and more, which are some of the main interests in MARL research. Even more importantly, they are designed in a way that even SOTA algorithms do not reliably solve them.
>
> [--] We understand that the reviewer feels the selected environments are far from real-world settings, and would prefer a benchmark that covers mostly robotic settings. However, such papers already exist [e.g. 5] and benchmark robotics simulations. This work is not meant to benchmark those, but instead focus on multi-agent interactions (agent coordination, multi-agent credit assignment, sub-optimal equilibriums, etc) that MARL is attempting to address. We hope the reviewer can recognise that research in such settings (even matrix games are still being studied) has been leading to continuous advances in the field for decades, and should not be discarded just because it does not directly relate to the physical world.
>
> [1] F. Christianos et al. Shared Experience Actor-critic for Multi-agent Reinforcement Learning. In 34th Conference on Neural Information Processing Systems (NeurIPS), 2020.
>
> [2] F. Christianos et al. Scaling Multi-agent Reinforcement Learning with Selective Parameter Sharing.  In International Conference on Machine Learning (ICML), 2021.
>
> [3] A. Rahman et al. Towards Open Ad Hoc Teamwork Using Graph-based Policy Learning. In International Conference on Machine Learning (ICML), 2021.
>
> [4] I. Ahmed et al. Towards Quantum-Secure Authentication and Key Agreement via Abstract Multi-Agent Interaction. International Conference on Practical Applications of Agents and Multi-Agent Systems (PAAMS), 2021.
>
> [5] Y. Duan et al. Benchmarking Deep Reinforcement Learning for Continuous Control. In International Conference on Machine Learning (ICML), 2016.

---

### Official Review · Reviewer_t79G · 2021-07-05

**Rating:** 7
**Confidence:** 3
**Clarity:** First half is more clear than the sec…

**Strengths:**

The paper considers an empirical evaluation of multi agent deep RL algorithms of different types and compares their performance on different environments; two of which are new to this paper or seem to have appeared previously in the literature but are newly integrated in a codebase with others. Given the complexity of the task, I find the first half of the paper nicely clear as a summary of differences between the algorithms and environments.

The paper's strengths in comprehensiveness lead naturally to its weaknesses in drawing structured conclusions from the empirical evaluation and emphasizing fundamental algorithmic or environment differences related to the math, rather than the phenomenology of specific simulations or environments.

**Weaknesses:**

Evaluation protocol 4.1: a bit more detail could be helpful here.
Section 4.3, performance metrics:
what about AUC; learning curves, or other summaries besides maximum and average requirements?
Fig. 1 is impossibly difficult to read. At least the legend could be made larger so it is legible.
Similarly, table 3 is also extremely difficult to read. Can the information be summarized further for the main text?
Statistical tests of fig. 5: multiple testing adjustments?
Section 5.3 parameter sharing:
Can you include more details on how the parameters were shared in the main text, how the update rules were coordinated and so on, concurrency and so on?

Overall: It could have been helpful to have more detailed comparisons of algorithms from the point of view of aspects of cooperation of the different environments. Some of this is gestured towards, e.g. lines 324-325 but this kind of structured comparison is most helpful.
Similarly section 332-348 would benefit from not just "phenomenological descriptions" of what happens in the experiments, e.g. " so that agents are not stuck in the narrow passages" but maybe with more specific descriptions of algorithm internals or how different algorithms succeed at these types of things.

Overall I appreciate the comprehensiveness of the evaluation but it is difficult for an outsider to glean conclusions from the extensive evaluation; more structured comparison, or organization in conclusions could be helpful here to draw on the extensive work done to provide crisper recommendations.

**Additional Feedback:**

The paper could be improved by more structured comparison of conclusions (I am not an expert on multi-agent RL so perhaps this would be more self-evident to someone working in the field).

**Correctness:**

To the best of my knowledge. There could be improvements with regards to statistical testing and more information on empirical performance evaluation.

**Documentation:**

There could be more information on the new environments in the main text.

**Relation To Prior Work:**

This paper seems to be a synthesis paper, e.g. building on prior work; while I appreciate the clarity of the summary, it may be clearer to precisely delineate what differs.

**Summary And Contributions:**

This paper develops a set of evaluations and comparisons of multi-agent RL algorithms. They also extend a previous codebase with a few additional algorithms and a redesigned api (for example allowing parameter sharing), and new environments.

The paper considers three algorithms for independent learning, centralized training decentralized execution, and value decomposition. There are 25 different learning tasks that are cooperative in the sense of shared reward signals; there is a taxonomy of difficulties in environments. Two matrix games follow previous work, and two new environments.

===
Thanks to the authors for the clarifications; I've raised my score.

---

> ### Author Response · Authors · 2021-07-14
> **Response to reviewer t79G**
>
> We would like to thank reviewer t79G for their review.
>
> We kindly ask the reviewer to specify which additional details might be of help in Section 4.1. We would gladly add any suggestions in the next/CR version of the paper. We tried to include all basic information for reproducing the experiments in Section 4, and we believe that combined with the Appendix and EPyMARL all the experiments can be fully reproduced.
>
> We have added more information about parameter sharing in Section 4.2.
>
> We present the learning curves in Appendix F.
>
> Regarding AUC, we present the Area Under the (Learning) Curve with respect to evaluation returns normalised by the number of evaluations in Tables 6 and 8. If the reviewer referred to a different AUC metric, we would appreciate any form of clarification.
>
> We also increased the size of the legend in Figure 1.
>
> We kindly ask the reviewer to further clarify their comment regarding Figure 5 since it shows visualisations of MPE tasks without any statistical tests. We would gladly add suitable comparisons in the next/CR version of the paper.
> We summarise Table 3 in Section 5, where we discuss the achieved returns of each algorithm in different environments.
>
> In Section 6, we build intuition and summarise main insight from our evaluation by tying properties of all considered algorithms and their classes to properties of learning environments and respectively achieved evaluation metrics. Additionally, researchers can use the EPyMARL codebase to train the nine algorithms (or extensions of these algorithms) in new environments.

---

### Official Review · Reviewer_ReJK · 2021-07-06
**Review of cooperative MARL benchmark**

**Rating:** 8
**Confidence:** 3

**Strengths:**

[S1] Extensive comparison. The number of compared algorithms and tasks is around the limit of what it is possible to compare in one paper with academic-level compute. The selection of algorithms and tasks also covers several important axes of variation for MARL, such as PG vs. TD, centralised vs. decentralised training, full vs. partial observability, etc.

[S2] The benchmark does in fact provide some generalisable insights in Section 6. None of these felt extremely surprising (e.g. indep. RL often works surprisingly well; centralised VFs help under partial observability), but there's value in providing empirically rigorous evidence for them nonetheless.

[S3] Writing is clear, hyperparametrs and experiment details were fairly thorough.

**Weaknesses:**

[W1] The proposed benchmark suite is mostly a collection of existing environments and re-implementations of existing algorithms (modulo the newly-released LBF benchmark, and the completely new RWARE benchmark). It therefore has limited novelty—I assume most of the value of the "benchmark" is in having a predefined suite for testing on in future?

[W2] The paper has an unusual mix of contributions, and it's not clear which are meant to be the primary ones. Is it meant to be an experimental paper that provides standalone insights, or a paper that proposes a new benchmark to beat? If the former, the findings seem somewhat underwhelming on their own. If the latter, how easy is it to run all the proposed algorithms on all the proposed environments? What about adding new algorithms or environments to the bake-offf? It seems like the environment code is split across a few different repositories, which might make this harder.

**Additional Feedback:**

Currently leaning accept on this. Results are not earth-shatting and the contributions are somewhat incremental in nature (a new environment here, an extension of an existing codebase there). However, the benchmark was well-executed, and I think this kind of work is valuable for creating common knowledge in the field about what approaches do or don't work.

------

(review last checked/updated 2021-07-19)

**Clarity:**

[CL1] What are the "cooperation settings" referred to on L154?

[CL2] How are returns being normalised in these figures?

**Correctness:**

[CO1] In Table 1, scores on the matrix games seem to be maxed out almost uniformly across all algorithms. This suggests that they are not challenging enough. It may be worth decreasing the number of time steps to make the benchmark more informative.

**Documentation:**

[D1] Documentation looked good for the new environments and EPyMARL. Note weakness [W2] above though—how hard is it to run the whole bake-off on a new algorithm? Is there any code dedicated to this?

**Ethics:**

No concerns.

**Relation To Prior Work:**

This is mostly a compilation of prior work into a new suite. Existing discussion therefore seems adequate.

**Summary And Contributions:**

This paper proposes a suite of environments (three preexisting, two novel) for cooperative MARL. It then evaluates a set of popular MARL algorithms across the suite, and draws tentative conclusions regarding the value of key algorithm design axes such as centralised vs. independent training, VDN vs. QMIX, etc.

---

> ### Author Response · Authors · 2021-07-14
> **Response to reviewer ReJK**
>
> We would like to thank reviewer ReJK for their review.
>
> [W1] The main contribution of this paper is the consistent evaluation of nine MARL algorithms in a diverse set of cooperative environments, two of which (LBF, RWARE) were newly developed for this study to test learning and coordination under sparse rewards and partial observability. To the best of our knowledge, there is no such extensive study in the literature. Our work can be used as a reference for MARL researchers and practitioners to guide their respective research by (among others)
> - informing the selection of appropriate baselines for MARL research
> - serving as a reference for the expected performance of evaluated algorithms
> - connecting the suitability of classes of MARL algorithms to properties of environments (such as partial observability and centralised critics)
>
> We also propose/maintain two new environments that can be used in future MARL research.
> The two proposed environments, even though they are grid-worlds, pose a significant challenge for modern MARL algorithms. They also complement the existing MARL environments and target different types of challenges as presented in Table 2.
>
>
> [W2 & D1]  Please see [W1] for more details about the scope of the paper.
> We have attempted to make the whole codebase easily installable while also retaining modularity. Researchers will be able to either use the codebase without the environments or perhaps the environments on their own (without specifically installing EPyMARL).
> That said, we will do our best to provide simple installation instructions to make sure researchers can even use the whole framework easily (full instructions found in the Appendix and the README file on the EPyMARL repository)
>
> We also added instructions for hyperparameter search and running the whole suite of environments (it involves running the `search.py` file which is included in the code repository).
>
> [CO1] From the experiments we observe that all algorithms converge to a sub-optimal risk-free local minimum policy in all matrix games, except penalty with k=0 (max episodic returns in all penalty games are 250). The matrix games are not maxed out but instead have proven to be much harder than anticipated for the tested algorithms.
>
> [CL1] 'Cooperation settings' refers to the number of agents that have to load a food item simultaneously. We have updated the revised version to clarify the meaning of 'cooperation' in this context, in Section 3.4.
>
> [CL2] We would like to thank the reviewer for noticing the missing formula. We include it in the revised version in Section 5.

---

> > ### Comment · Reviewer_ReJK · 2021-07-19
> > **Changing score**
> >
> > Thank you for the thorough response. Your response to [W1] has helped me understand what the paper is actually trying to achieve, and after taking a closer look at the Github repo I'm less concerned about [W2/D1]. [CO1] was a misunderstanding on my part—I assumed it was maxed out because they all had the same return. I'm going to up my score for the paper accordingly.

---

### Official Review · Reviewer_AMzt · 2021-07-06
**The new environments look useful, the libraries are well packaged, and the experiments appear useful and trustworthy.**

**Rating:** 8
**Confidence:** 3
**Clarity:** The paper is clear and well written.

**Strengths:**

The first major strength of this paper is that it contains multiple useful tools, which are all reasonably licensed and packaged.
The experiments are also strong, since they used an appropriate hyper-parameter search, statistical analysis, and explanations for differences found between algorithms.

**Weaknesses:**

The only notable limitations of this work is the depth of analysis provided for any particular algorithm. However, given the scope of the paper, this is completely understandable.

**Additional Feedback:**

I have no additional feedback or respond to provide.

**Correctness:**

The claims in this submission appear to be correct. The evaluation and experiment design are appropriate.

**Documentation:**

The benchmark is licensed in an appropriate way for research uses, and a plan appears to be in place for future hosting and maintenance. The appendix contains sufficient information to reproduce all results presented.

**Ethics:**

I do not believe there are any ethical concerns that warrant further discussion or review.

**Relation To Prior Work:**

The clearly indicate what work is new work of their own. However, I'm not sufficiently familiar with MARL to comment on its coverage of prior work.

**Summary And Contributions:**

This paper makes several contributions.
Firstly, it includes an extension of PyMARL to handle more algorithms and allow tuning more details of those algorithms.
Secondly, it includes two new MARL environments.
Thirdly, it includes experimental results comparing most of the commonly used algorithms in this field, as well as analysis of those algorithms' performances.
The resulting experiments show that the new environments provide unique challenges compared to previously available benchmarks.
A hyper-parameter search was used to tune the algorithms for each benchmark, and statistical analyses are used to indicate which found results were significant.

---

> ### Author Response · Authors · 2021-07-14
> **Response to reviewer AMzt**
>
> We would like to thank reviewer AMzt for their review. We are glad that they liked our paper.

---

### Official Review · Reviewer_PyTY · 2021-07-06
**An exhaustive evaluation of Multi-Agent RL Methods**

**Rating:** 7
**Confidence:** 3
**Correctness:** Yes.
**Clarity:** Yes.

**Strengths:**

The evaluations are thorough and it's nice that a confidence interval is computed for the evaluations.

**Weaknesses:**

None that stand out.

**Additional Feedback:**

N/A

**Documentation:**

Yes, there is a maintenance plan described.

**Ethics:**

No.

**Relation To Prior Work:**

Yes.

**Summary And Contributions:**

This work does a benchmark evaluation of multi-agent RL methods across a number of tasks.

---

> ### Author Response · Authors · 2021-07-14
> **Response to reviewer PyTY**
>
> We would like to thank reviewer PyTY for their review. We are glad that they liked our paper.

---

### Author Response · Authors · 2021-07-14
**General Comment**

We would like to thank all reviewers for their reviews. We have uploaded a revised version of the paper to address some of the comments of the reviewers. More specifically:
- We have added more information to address the comments regarding clarity of reviewer ReJK, in Sections 3.4 ([CL1]) and 5 ([CL2]).
- We have added information about parameter sharing as suggested by reviewer t79G in Section 4.2.
- We have increased the font size of Figure 1 as suggested by reviewer t79G.
- We have performed statistical tests in Table 7.

---

### Decision · Program_Chairs · 2021-07-26

**Decision:**

Accept

**Comment:**

This paper has multiple contributions. Firstly it extends the existing PyMARL, which is called as EPyMARL (Extended
PyMARL) to implement more algorithms and allow for flexible configuration of different implementation details. Second, the paper implemented and open-sourced two new multi-agent environments: Level-Based Foraging (LBF) and Multi-Robot Warehouse (RWARE). Lastly it provides an extensive study of different MARL algorithms on various MARL environments.

The paper focuses on addressing important issue in MARL: the lack of clear benchmark and evaluation of the MARL agents across a diverse set of MARL environments. The reviewers in general found the contributions of the paper meaningful. The authors did a good job in the rebuttal. Thus, this paper deserves to be presented as a poster at NeurIPS 2021 Datasets and Benchmarks Track.